# Cancer Biogenesis in Ribosomopathies

**DOI:** 10.3390/cells8030229

**Published:** 2019-03-11

**Authors:** Sergey O. Sulima, Kim R. Kampen, Kim De Keersmaecker

**Affiliations:** Department of Oncology, KU Leuven, LKI–Leuven Cancer Institute, 3000 Leuven, Belgium; sergey.o.sulima@gmail.com (S.O.S.); kim.kampen@kuleuven.be (K.R.K.)

**Keywords:** ribosome, cancer, ribosomopathies, translational fidelity

## Abstract

Ribosomopathies are congenital diseases with defects in ribosome assembly and are characterized by elevated cancer risks. Additionally, somatic mutations in ribosomal proteins have recently been linked to a variety of cancers. Despite a clear correlation between ribosome defects and cancer, the molecular mechanisms by which these defects promote tumorigenesis are unclear. In this review, we focus on the emerging mechanisms that link ribosomal defects in ribosomopathies to cancer progression. This includes functional “onco-specialization” of mutant ribosomes, extra-ribosomal consequences of mutations in ribosomal proteins and ribosome assembly factors, and effects of ribosomal mutations on cellular stress and metabolism. We integrate some of these recent findings in a single model that can partially explain the paradoxical transition from hypo- to hyperproliferation phenotypes, as observed in ribosomopathies. Finally, we discuss the current and potential strategies, and the associated challenges for therapeutic intervention in ribosome-mutant diseases.

## 1. Introduction

Precise conversion of genomic information into functional proteins is central to life. This is accomplished by the largest and most abundant ribozyme in the cell—the ribosome. Comprising ~6000 ribosomal RNA (rRNA) bases and 80 ribosomal proteins (RPs), each ribosome must be assembled in a complicated and energetically demanding process taking place in both the nucleus and cytoplasm of cells. Accurate sequential ribosome assembly and subsequent quality checks are crucial early components of the ribosomal lifecycle, and it has been estimated that thousands of ribosomes are being constructed and functionally checked every minute in a growing eukaryotic cell [1]. 

Impaired ribosome biogenesis and function is the underlying cause of diseases called ribosomopathies [2]. These disorders are characterized by a wide spectrum of symptoms which, due to a shortage of functional ribosomes, broadly fall under the category of cellular hypo-proliferation phenotypes. For example, bone marrow failure and anemia are seen in many ribosomopathies. Whereas such hypo-proliferation phenotypes were deadly in the past, supportive care in the form of steroids, red blood cell transfusions, and bone marrow transplants now allows patients to survive this initial disease phase. Paradoxically, many of these diseases are characterized by an elevated risk to progress to a hyper-proliferative cellular state and ultimately cancer later in life [3]. How can these ailments first appear as diseases founded on a lack of cell proliferation, but later turn into cancer—a disease of uncontrolled growth? William Dameshek, the founding editor-in-chief of the journal Blood, introduced this paradox back in a 1967 editorial [4]. Here, before the molecular genetics era, Dameshek described the observation that patients who initially develop a hypo-proliferative disease, such as aplastic anemia, tend to be at higher risk of hyper-proliferative diseases, such as acute leukemia. Ribosomopathies are thus examples of the longstanding and unsolved “Dameshek’s Riddle”.

The involvement of ribosomal mutations in cancer has been further demonstrated by the recent identification of somatic RP mutations in a variety of hematopoietic and solid tumors [5]. Many of these mutations also cause ribosome assembly and proliferation defects in human and mouse cell models, suggesting that RP defects might fulfill functionally similar roles in congenital and somatic disease. Such somatic RP-defective cancer models can thus provide valuable insights to elucidate the contribution of ribosomal mutations on cancer progression. Several comprehensive reviews have recently provided excellent overviews of the spectrum of ribosome defects in cancer, and have highlighted the increasing interest in the connection between ribosomal defects and oncogenic processes [5,6,7,8]. In the current review, we extend on those topics and summarize and comment on the emerging data that are paving a path towards understanding the cancer predisposition nature of ribosomopathies.

## 2. Ribosome Biogenesis and Ribosomopathies

### 2.1. Overview of Ribosome Assembly

In prokaryotes, ribosome biogenesis is determined and driven purely by the intrinsic properties of its rRNA and RPs, and follows a well-established assembly pipeline [9]. Ribosome assembly in eukaryotes, however, is much more complex and less well understood (recently reviewed in [10]). While the secondary and tertiary structures of the ribosomal core are well conserved across species, eukaryotic ribosomes are much larger and more complex, containing additional RNA expansion segments as well as many additional proteins or protein extensions [11]. As a result, their construction is fundamentally different, and these structural and functional differences are the basis for approximately half of the current arsenal of antibiotics (reviewed in [12]). Additionally, over 200 accessory trans-acting factors guide the eukaryotic ribosome assembly process, but do not become part of the mature ribosome structure. These proteins include GTPases, ATPases, kinases, helicases, and chaperones, and promote essential functions such as rRNA processing, modification, and folding. Finally, eukaryotes also bear the added burden of exporting immature ribosomal subunits from the nucleus to the cytoplasm. 

The ribosome assembly process (summarized in Figure 1) begins in the nucleolus, where most of the rRNA molecules are transcribed as precursor rRNA. These pre-rRNAs undergo a series of cleavages and trimmings, along with extensive modifications by ~70 small nucleolar RNAs and protein co-factors, to become the mature 18S, 25S, 5.8S, and 5S rRNAs. Ribosomal protein messenger RNAs (mRNAs) are transcribed in the nucleoplasm, translated in the cytoplasm, after which they are transported back to the nucleolus for participation in subsequent ribosome assembly steps. These steps consist of the formation of numerous ribosome assembly intermediates and the export of pre-40S and pre-60S particles from the nucleolus into the nucleus and subsequently to the cytoplasm for final rRNA processing and protein associations. The fully assembled and mature large 60S and small 40S subunits can then interact to form translationally active 80S ribosomes.

Ribosome biogenesis is tightly controlled through surveillance systems that ensure the functionality of newly assembled ribosomal subunits. Specifically, pre-40S subunits undergo a proofreading step at the end of their cytoplasmic maturation [13]. This involves a translation-like cycle, whereby mature 60S subunits join pre-40S subunits in complex with translation initiation factors. However, the resulting 80S-like ribosomes are not translationally active, as they do not contain mRNA or transfer RNA (tRNA). Rather, this translation-like cycle serves as a final quality checkpoint in which major functions of the maturing small subunit, such as binding to 60S subunits and to translation factors, are examined. In an analogous “test drive” of immature 60S subunits, the ability to bind biogenesis ligands that structurally mimic translation factors as well as the ability to undergo GTP hydrolysis is being functionally tested [14,15]. These mechanisms provide primary quality control checks on assembly and function of the ribosomal subunits before they are released into the pool of actively translating ribosomes.

Almost the entire cellular proteome is synthesized by cytoplasmic ribosomes, with some ribosomes directed to the endoplasmic reticulum (ER). However, a small complement of proteins required for oxidative phosphorylation is translated exclusively by mitochondrial ribosomes. These mito-ribosomes, consisting of 28S and 39S subunits composed of mitochondrial genome-encoded mito-RPs and rRNA, structurally more closely resemble prokaryotic ribosomes. While mito-ribosomes are less well understood, the importance of further investigating mitochondrial translation is illustrated by the fact that mutations in mitochondrial RP genes are associated with mitochondrial dysfunction disorders (reviewed in [16]).

### 2.2. Ribosomopathies: Manifestations of Ribosome Assembly Defects

Given the central role of ribosomes in every cell, defects in ribosome assembly and/or function can cause cellular malfunction and diseases. Ribosomopathies can be defined as any disease associated with a mutation in a ribosomal protein or biogenesis factor impairing ribosome biogenesis, in which a defect in ribosome biogenesis or function can be clearly linked to disease causality [3]. In 1999, recurrent mutations in the ribosomal protein gene *RPS19* (also known as eS19) were reported in patients with Diamond–Blackfan anemia (DBA), a congenital bone marrow failure syndrome [17]. Since then, mutations in a number of RPs have been identified in up to 50% of patients with DBA [18]. Moreover, other congenital syndromes have been linked to non-RP related ribosomal defects, with the most studied examples being Schwachman–Diamond syndrome (SDS), X-linked dyskeratosis congenita (DC), cartilage hair hypoplasia (CHH), and Treacher Collins syndrome (TCS) [2]. In addition, haplo-insufficiency for *RPS14* (uS14) in 5q−myelodysplastic syndrome (5q-MDS) leads to an erythroid differentiation defect highly similar to DBA. This disorder is characterized by a somatically acquired deletion of the entire 5q chromosome, and other genes in the deleted region may therefore also contribute to the disease phenotype [19,20]. Rarer ribosomopathies, with incidences of less than 1 in 200,000, include isolated congenital asplenia and North American Indian childhood cirrhosis. 

Each of the genetic abnormalities in these disorders disrupts a specific step in ribosome biogenesis, overviewed in Figure 1. For example, approximately 90% of SDS cases are caused by mutations in the *Shwachman-Bodian-Diamond Syndrome* (*SBDS*) gene, leading to the loss of SBDS protein expression [21]. This trans-acting factor is involved in a late step in the cytoplasmic maturation of 60S subunits by promoting the release of eukaryotic initiation factor 6 (EIF6) from pre-60S subunits [22]. EIF6, also known as the “anti-association factor”, keeps the nascent 60S subunit in a functionally inactive state during cytoplasmic 60S assembly, but needs to be released for final 60S maturation and association with the 40S subunit. In *SBDS*-mutant SDS patients, cells are unable to efficiently release EIF6, thereby stalling 60S maturation [23]. Moreover, CHH is caused by mutations in *RNA component of Mitochondrial RNA Processing endoribonuclease* (*RMRP*), a long non-coding RNA component of the RNase MRP complex. *RMRP* mutations or knock-down affect rRNA processing by inhibiting cleavage of pre-rRNA in the internal transcribed spacer 1, leading to reduced levels of mature 18S and 5.8S rRNAs [24,25]. Additionally, mutations in several RPs in both subunits have a direct impact on pre-rRNA processing in DBA. The diverse defects in rRNA processing that have been described in DBA cells can even be exploited to rapidly diagnose DBA patients [26,27,28].

As observed in most ribosomopathies, a shortage of mature ribosomes causes hypo-proliferative clinical symptoms such as anemia, bone marrow failure, and dysostosis. Additionally, disrupted ribosome assembly increases the availability of free RPs, which can activate TP53 and further augment the hypo-proliferative phenotypes. Intriguingly however, these diseases frequently transition to a hyper-proliferative state later in life, and ribosomopathy patients are at significantly higher risk to develop various types of cancers. Generally, ribosomopathy patients have a 2.5 to 8.5-fold higher risk of developing cancer throughout their lifetimes. However, for particular cancer types, these risks can be up to 200-fold higher (Figure 2) [29,30,31] and rise even further after hematopoietic stem cell transplantation [30]. How initially too few cells can ultimately turn to too many is an important open question, and the potential mechanisms of this transition are discussed below.

## 3. Oncogenic Mechanisms in Ribosomopathies

A number of promising mechanistic explanations for the paradoxical transition from hypo-proliferation to cancer have recently emerged, which can be broadly classified into three categories. The first category concerns the direct effect of ribosomal gene mutations or deletions on ribosomal function. The resulting defects not only lead to ribosome insufficiency due to ribosome misassembly, but also to altered translation carried out by the misassembled, structurally unique ribosomes. This altered translation can manifest, for instance, as a specific translatome that is shifted towards growth-promoting and oncogenic protein expression signatures. Secondly, the extra-ribosomal function of some RPs involved in ribosomopathies might also be relevant to promoting the oncogenic action of ribosome defects, as some of these functions relate to regulation of major cancer genes such as *TP53* and *MYC*. The third category highlights the influence of ribosome defects on various cellular stress conditions, which can favor a specific translational program as well as promote the acquisition of potentially rescuing mutations. We discuss emerging insights regarding these three partially overlapping cancer-promoting paths in more detail in the following sections.

### 3.1. Ribosomal Functions

Historically, only one type of ribosome was assumed to exist in all cells and tissues of the body, passively translating all mRNAs equally. An alternate system was demonstrated in 1987, albeit artificially, when mutated ribosomes favoring specific translation of a single mRNA species in *Escherichia coli* were generated, giving rise to the idea of “specialized” ribosomes [32]. Several decades later, ribosomal heterogeneity observed both at the level of core RPs and rRNA and of proteins interacting with the ribosome suggest that specialized ribosomes also exist naturally (reviewed in [33,34]). Importantly, differences in ribosome composition facilitate specialized functions, as such diverse ribosomes display mRNA-specific protein synthesis. Along the same lines, a number of recent studies on congenital and somatic RP mutations in ribosomopathies and cancer support that, in addition to negatively impacting ribosome assembly, these mutations also influence ribosomal function. The resulting altered protein expression programs could be instrumental for ultimately enabling a pre-oncogenic state, and we discuss various facets of such ribosomal “onco-specialization” below. 

#### Specialization through Altered Translation Potential

Speed and accuracy are critical properties of translation. The ribosome synthesizes proteins at a speed of 15–20 amino acids/second with an error rate of 10^−3^–10^−4^/codon [35]. Many RP-mutant cell models display altered translational speed and fidelity, suggesting that such defects might also influence the cellular state in ribosomopathies. In this context, *RPS23* (uS12)-mutant ribosomes were shown to promote dysmorphism in a recently described ribosomopathy via increased levels of amino acid misincorporation, without affecting the general rates of translation [36]. While this rare disease has not yet been linked with increased cancer risks, decoding-defective ribosomes in other ribosomopathies may provide novel insights into the cellular consequences of altered translation. 

Another example of translational fidelity includes the interaction of ribosomes with various mRNA secondary structures. Internal ribosomal entry site (IRES) elements are examples of such mRNA regulons, which can recruit ribosomes independent of the canonical 5’ cap-driven translation initiation. These elements are frequently found on mRNAs encoding stress-response genes, enabling rapid activation of their translation in stress conditions when 5’ cap-dependent translation is inhibited. It is becoming clear that mutations in ribosomes can influence the efficiency of IRES-mediated translation. For example, the leukemia-associated R98S mutation in *RPL10* (uL16) drives specific and constitutive IRES-mediated overexpression of the anti-apoptotic factor BCL-2. This enables ribosome-mutant cells to survive chemotherapy induced cell death as well as high levels of oxidative stress associated with the *RPL10-R98S* (uL16-R98S) mutation [37]. This cellular survival advantage, endowed by the specialized function of RPL10-R98S (uL16-R98S) mutant ribosomes, facilitates a pre-oncogenic state. In regards to ribosomopathies, mutations in *RPS19* (eS19) and *RPL11* (uL5) observed in DBA reduce the IRES-mediated translation of erythroid differentiation factors *BAG1* and *CSDE1* in mouse models of disease as well as in DBA patient samples [38].

Besides RP mutations, altered RNA modification can also influence IRES-mediated translation. Ribose 2′-hydroxyl methylation and pseudouridylation are the most common types of rRNA modifications [39]. *DKC1*, which encodes the enzyme dyskerin responsible for pseudouridylation, is mutated in DC. These mutations prevent the translation of some IRES-containing mRNAs, such as the tumor suppressor genes *TP53* and *CDKN1B* and the anti-apoptotic factors *BCL2L1* and *XIAP* [40,41]. Remarkably, *DKC1* mutations increase the translation of other IRES-containing mRNAs, such as the angiogenic factor *VEGF* [42], underscoring the diversity of IRES elements. These examples highlight that altered IRES-dependent translation by defective ribosomes can contribute to cancer progression of ribosomopathies by favoring the expression of pro-oncogenic proteins and/or disfavoring tumor suppressor proteins. 

Another class of cis-acting mRNA control elements include programmed -1 ribosomal frameshift (-1 PRF) signals. Such signals cause translating ribosomes to “slip” backwards by one nucleotide on an mRNA, leading to continued translation in a new reading frame and termination at a premature stop codon [43]. Higher rates of -1 PRF on an mRNA are therefore associated with lower rates of protein expression of that mRNA. Ribosome-associated mutations can significantly alter -1 PRF rates. For example, ribosomes carrying a mutation found in leukemia display lower rates of -1 PRF at specific -1 PRF signals in genes of the JAK-STAT signaling cascade, which is associated with overexpression of JAK-STAT proteins in ribosome-mutant leukemia cells [44]. Furthermore, depletion of DKC1 broadly reduces translational fidelity and -1 PRF in both yeast and human cells [45].

Additional recently discovered mRNA regulons such as RNA G-quadruplex structures, 5′ terminal oligopyrimidine tract motifs, the translation inhibitor element, pyrimidine-rich translational element, and cytosine-enriched regulator of translation, are believed to interface with the translational apparatus on distinct pro-tumorigenic mRNAs (reviewed in [46]). We hypothesize that messages containing such secondary structures might be differentially translated by ribosomes in ribosomopathies, further broadening the repertoire of their specialized function and potential to influence cancer progression. 

Additional lines of evidence support the hypothesis that certain phenotypes associated with RP and biogenesis factor defects are caused by specific changes in translation rates. For example, the *Rps19* (eS19) and *Rpl11* (uL5) mutant zebrafish lines show a decrease in globin translation in erythroid cells [47]; reduced expression of RPS19 (eS19), RPL5 (uL18), RPL11 (uL5), or RPS24 (eS24) in DBA cells leads to a specific translational reduction of the master regulator of hematopoiesis GATA1 [48]; and cells derived from nerve sheath tumors that developed in 17 different heterozygous RP gene mutant zebrafish lines all displayed a specific defect in Tp53 translation [49]. 

### 3.2. Ribosome-Independent Functions

Because some ribosomopathies are caused by defects in a ribosome biogenesis factor, and because some RPs have extra-ribosomal functions, it is important to consider ribosome-independent mechanisms as a potential source of oncogenesis in these diseases. In regards to the former, *SBDS* mutations in SDS were found to specifically affect the capacity to translate C/EBP α and β isoforms, important regulators of granulocyte differentiation. SBDS function is specifically required for efficient translation re-initiation into the protein isoforms, which is controlled by a single cis-regulatory upstream open reading frame (uORF) in the 5’ untranslated regions (5’ UTRs) of both mRNAs [50]. The inability to carry out proper C/EBP translation may explain the impaired hematopoiesis and leukemia predisposition observed in SDS patients. 

A large number of extra-ribosomal moonlighting functions of RPs have been described, of which many relate to established oncogenes and tumor suppressors. We refer to other specialized reviews for an extensive overview on this topic [51], and will only focus on the most established extra-ribosomal functions here. 

Several RPs affected in ribosomopathies have an extra-ribosomal function involving a negative feedback loop with c-MYC. This factor enhances ribosome biogenesis by inducing both rRNA and RP transcription, and certain RPs in turn inhibit c-MYC levels and function [52]. RPL11 (uL5) binds c-MYC at promoter regions of c-MYC target genes, inhibiting c-MYC-dependent transcription [52,53]. In addition, RPL5 (uL18) and RPL11 (uL5) jointly bind to the c-MYC mRNA and guide it to the RNA-induced silencing complex (RISC) for degradation [54]. A similar mechanism has been described for RPS14 (uS11) [55]. These studies suggest that mutations/deletions in these RPs might lead to oncogenic c-MYC overexpression. In support of this notion, RPL11 (uL5) and RPL22 (eL22) inactivation shortens the latency of lymphoma development in mouse models and is associated with c-MYC upregulation [56,57]. 

Another well-established extra-ribosomal function of some RPs relates to TP53 regulation. The causes of the hypo-proliferative clinical symptoms of most ribosomopathies have long been linked to this tumor suppressor [58]. However, TP53 does not provide the only connection between ribosome biogenesis and cell cycle regulation, and TP53 independent mechanisms of cell-cycle arrest after ribosomal stress have been described [59]. TP53 plays a fundamental role in the surveillance of protein translation and can be activated by ribosome dysfunction. In particular, the MDM2 protein is a central regulator of TP53, acting as an ubiquitin ligase that leads to TP53 degradation. Ribosome assembly defects result in freely available RPs, some of which (e.g. RPL5 (uL18)/RPL11 (uL5)) can bind and sequester MDM2, preventing MDM2-induced TP53 degradation and thereby inducing TP53 activity [27,60]. Consistent with this, *RPL5* (uL18) and *RPL11* (uL5) mutations in DBA cells are characterized by defects in both ribosome biogenesis and cell cycle progression [27,54,60]. Collectively this suggests that loss of the tumor suppressive functions of RPL5 (uL18) or RPL11 (uL5) via inactivating mutations might predispose DBA patients to cancer development. While this extra-ribosomal function is most established for RPL5 (uL18) and RPL11 (uL5), many other RPs have also been described to regulate TP53 by binding MDM2 [51].

Additionally, TP53 has a well-characterized role in inhibiting the RNA Pol I transcription machinery to repress rRNA synthesis [61]. Loss of TP53 in cancer would therefore provide a further mechanism for bypassing these growth-inhibiting control mechanisms. Consistent with this idea, a recent study described inactivating lesions in TP53 as early events required for the transition of SDS to acute myeloid leukemia (AML) [62]. The concept of ribosomal lesions broadly acting as mediators for additional mutagenesis represents a novel developing concept, which we discuss in the following section. 

### 3.3. Cellular Stress and Metabolic States

Lastly, the role of misassembled/mutant ribosomes on the transition to cancer might also be more indirect. In addition to inducing specific translational changes, ribosomal lesions might shape the right conditions—by promoting cellular stress—which favor the appearance of additional, rescuing mutations. In particular, it is becoming clear that RP-mutant diseases suffer from high cellular oxidative stress due to increased levels of reactive oxygen species (ROS). The mechanisms by which defective ribosomes lead to elevated ROS are poorly understood. In the case of the leukemia associated *RPL10-R98S* (uL16-R98S) mutation, enhanced levels of ROS may arise from increased peroxisome activity. Peroxisomes are cellular organelles in which oxidation of long-chain fatty acids occurs, resulting in the production of high levels of hydrogen peroxide (H_2_O_2_). Several peroxisomal enzymes, such as PAOX, are transcriptionally upregulated in *RPL10-R98S* (uL16-R98S) cells [37]. By what mechanisms mutant ribosomes drive peroxisomal oxidation is unclear. Wild-type *RPL10* (uL16) has also been described to regulate the expression of proteins related to ROS production and to control mitochondrial ROS production in pancreatic cancer [63]. It is therefore possible that RPL10-R98S (uL16-R98S) is no longer able to properly perform these ROS regulatory functions, but this requires further research. Other ribosomal proteins have also been involved in oxidative stress responses. For example, ROS-inducing agents can cause RPS3 (uS3) to translocate to the mitochondria, where it can protect the cells from ROS-induced mitochondrial DNA damage [64]. Elevation of ROS levels can initiate different cellular outcomes, depending on the levels that are achieved. At low levels, cellular ROS levels can stimulate proliferation by activating the PI3K and MAPK signaling pathways [65,66,67,68]. At higher levels, ROS-associated oxidative stress becomes toxic and rather inhibits proliferation. In the context of ribosomal protein mutations, the latter seems to be the case: reduction of cellular ROS levels by means of an anti-oxidant can rescue the proliferation defects in *RPL10-R98S* cells [37], supporting that RPL10-R98S-associated ROS production impairs cell proliferation. Elevated ROS levels have also been linked to increased DNA damage and genomic instability. The *RPL10-R98S* (uL16-R98S) mutation was shown to be associated with ROS-mediated oxidative stress and elevated DNA damage [37,69]. Models of DBA also present enhanced oxidative stress and DNA damage: RPL5 (uL18) and RPS19 (eS19)-deficient mouse erythroleukemia-DBA clones and DBA patient samples display increased ROS levels, with higher levels of γH2A.X stained double strand DNA breaks and 8-oxoguanine oxidative DNA damage [70]. Additionally, SDS patient lymphocytes and an SDS mouse model showed elevated oxidative stress [71,72], and activation of DNA damage responses in hematopoietic cells from this mouse model was demonstrated [72]. In human HeLa cells, transient suppression of DKC1 induced oxidative stress accompanied by elevation of anti-oxidant enzymes [45]. While no changes in γH2A.X foci were observed, DNA lesions defined by protein poly ADP-ribosylation were increased. Finally, CHH patient samples display 2.4-fold higher expression of the ROS scavenger catalase [73], pointing towards a potential elevation of oxidative stress in this disease as well. Upregulation of ROS scavengers is however not a generalized phenotype in ribosomopathies: expression of genes involved in ROS detoxification, such as superoxide dismutase 2, is decreased in DBA models such as *Rpl11* (uL5) zebrafish mutants and *RPS19 (eS19)*-deficient human cell lines [74,75]. The detected increases in ROS levels may thus instead result from a reduced cellular detoxification capacity. However, enhanced production of ROS has also been proposed, e.g., due to elevated ER and mitochondrial stress [71] or enhanced activity of the peroxisome [37].

High oxidative stress can in turn induce mitochondrial dysfunction, interfering with adequate oxidative phosphorylation, mitochondrial respiration and consequently ATP generation [37,71]. This can on one hand contribute to the hypo-proliferative phenotypes observed in RP-mutant cancers and ribosomopathies [37,71,76]. On the other hand, increased DNA damage due to high oxidation can enable increased mutagenesis, empowering these cells to acquire rescuing mutations. In support of this, mutational signature analysis showed that the mutational patterns in *RPL10-R98S* (uL16-R98S) T-cell acute lymphoblastic leukemia (T-ALL) are dominated by C:G>A:T transversion mutations, which are typically caused by oxidative damage. Moreover, RP-mutant T-ALL and chronic lymphocytic leukemia (CLL) patients display higher mutational burdens compared to patients with wild-type ribosomes. These higher mutational loads are enriched for mutations that can reduce oxidative stress, such as *NOTCH1*-activating mutations in T-ALL and *TP53*-inactivating mutations in CLL [69,77]. Moreover, *TP53* mutations in patients with SDS have been described as early events in the transformation to AML [62]. According to these data, ribosomal lesions might function as a main source of the cellular mutagenic potential: they initiate high cellular stress levels which can elicit particular cellular responses such as the acquisition of and selection for rescuing mutations (Figure 3). Those cells that can survive the initial stress conditions and adapt ultimately emerge stronger and fitter. Furthermore, ribosomal RNA is a target for oxidative nucleobase damage, and increased oxidative stress levels can interfere with ribosome assembly and different sub-steps of the translation elongation cycle [78], as well as reduce the fidelity of protein translation [79]. This can, in turn, further augment ROS production, leading to an oxidative and translation-defective cellular “snowball” effect that might even further enhance a mutagenic phenotype.

This model might also in part explain the tissue-specificity paradox observed in ribosomopathies. Patients with mutations in ribosomal proteins show remarkably specific phenotypes, suggesting that ribosomal proteins have unique functions in different tissues, further adding to the concept of specialized ribosomes discussed above. However, most ribosomopathies also display impaired hematopoiesis, and several explanations accounting for this hema-centrism have been proposed [80,81]. Interestingly, only ribosomopathy patients who present with defects in hematopoiesis progress to cancer. For example, anemia and bone marrow failure are observed in DBA, DC, SDS, and CHH, each caused by mutations in a different ribosomal protein or assembly factor, and associated with increased progression to various cancers, including many hematologic malignancies (Figure 2). In contrast, Treacher Collins syndrome does not display any hematopoietic abnormalities and is not correlated with increased cancer risk. A possible explanation for the susceptibility of these disorders towards cancerous transformation is the increased sensitivity of the hematopoietic system to oxidative stress. Indeed, ROS have been directly linked to genetic instability in hematopoietic stem cells [82], which can increase the mutagenic pool and hence the likelihood of acquiring growth-promoting mutations. We speculate that *TCOF1* mutations in TCS may not induce oxidative stress, which could explain why there is no elevated tumor risk in this disease. It would therefore be of interest to investigate ROS levels in TCS in the future. 

In addition to elevated ROS, broad metabolic reprogramming is a general hallmark of cancer, as transformed cells depend on augmented nucleotide and protein synthesis and ATP production to fuel these processes. To meet the requirements of a cell during the transition from the hypo-proliferative to the hyper-proliferative phase, ribosome defective cells likely also rely on metabolic rewiring. A tight connection between ribosomes and cell metabolic regulation is also supported by the recent observation that the interaction partners of the ribosome, the so-called ribo-interactome, is enriched for proteins with a role in cell metabolism [83]. Nevertheless, not much is known about metabolic changes in ribosomopathies, and even less about their contribution to oncogenic transformation. However, glycolytic changes have recently been linked to ribosome defects. Microarray gene expression analysis on leukocytes from CHH patients revealed an upregulation of glycolysis enzymes, such as fructose-1,6-bisphosphatase 1 (FBP1), glucokinase (GK), and hexokinase 2 (HK2) [73]. SDS patient lymphoblasts presented low levels of pyruvate and impaired oxidative phosphorylation, accompanied by increased lactate levels, indicating that SBDS deficient cells are highly glycolytic [71]. It has been established that cancer cell derived lactate blocks proper immune cell function and surveillance [84]. Whereas CHH and SDS have thus been linked to elevated glycolytic activity, the opposite holds true for DBA: *Rpl11* (uL5)-deficient zebrafish and *RPS19* (eS19)-deficient mouse fetal liver cells downregulate genes encoding glycolytic enzymes and upregulate genes involved in aerobic respiration [75]. Interestingly, the key glycolytic enzymes pyruvate kinase isozyme 2 (PKM2), fructose-bisphosphate aldolase A (ALDOA), and lactate dehydrogenase A (LDHA) directly bind to ribosomes [83], making it tempting to speculate that altered ribosome availability in ribosomopathies may directly impact glycolytic enzyme availability and activity. 

Modulation of serine synthesis, a glycolysis-diverting pathway, has also been described in ribosome-mutant disorders. The proteome changes induced by the leukemia-associated *RPL10-R98S* (uL16-R98S) mutation were shown to mainly belong to metabolism pathways. In particular, combined transcriptome and translatome analysis revealed that one of the key enzymes in serine synthesis—phosphoserine phosphatase (PSPH)—is more efficiently transcribed and translated in *RPL10-R98S* (uL16-R98S) mutant cells. The resulting elevated serine is converted to glycine, a reaction in which one carbon is released and converted into formate, which in turn sustains nucleotide synthesis [85]. Similarly, fibroblasts of DBA patients present elevated levels of serine/glycine synthesis enzymes phosphoglycerate dehydrogenase (PHGDH), phosphoserine aminotransferase 1 (PSAT1), and the mitochondrial serine hydroxymethyltransferase 2 (SHMT2) [86]. Interestingly, a recent study highlights the requirement of SHMT2-driven serine catabolism for maintaining formylmethionyl-tRNAs, associated with proper mitochondrial translation initiation [87]. These data indicate that also in DBA, serine catabolism may donate carbons to formate to sustain increased nucleotide synthesis. 

Overall, several studies thus support that ribosome-defective cells undergo metabolic reprogramming to benefit from glycolysis and one carbon metabolism. These intriguing observations however require further investigation.

## 4. Conclusions and Therapeutic Potential

In the paragraphs above, several cellular changes induced by ribosome defects were discussed. In the following paragraphs, we attempt to put these observations together to construct a model that may explain the paradoxical transition from hypo- to hyperproliferative disease phenotypes in ribosomopathies (Figure 3). On one hand, the presence of a ribosome defect alters the functioning of the ribosome. The resulting “specialized ribosome” translates an altered protein output as compared to wild-type ribosomes, entailing overexpression of oncogenes such as BCL2, JAK-STAT proteins, or VEGF, and underexpression of tumor suppressors such as TP53 or CDKN1B. Altered extra-ribosomal regulation of proteins such as TP53 or MYC represents a second means by which cancer associated proteins can be affected. Dysregulation of the cellular oncogene/tumor suppressor balance is however not the only consequence. Occurrence of ribosomal lesions induces extensive oxidative stress and the generation of high levels of ROS. This is highly toxic to cells and impairs cellular growth. Indeed, in the context of the RPL10-R98S (uL16-R98S) mutation, reduction of ROS levels has been shown to completely rescue the proliferation defect [37,69]. Thus, as long as cells are unable to reduce ROS levels, hypoproliferation continues. ROS damages multiple cellular components, including the DNA, causing high mutation rates. At a certain moment, some of these mutations act as an anti-oxidant and lower the ROS levels, at which point transition to hyper-proliferation can occur: the oxidative stress induced block on cellular fitness is removed, allowing the disturbed oncogene/tumor suppressor balance to stimulate proliferation. Other mutations due to extensive DNA damage that further promote cell growth will be selected, strengthening the hyper-proliferation phenotype (Figure 3). 

Ribosomes are now seen as important contributors to cancer progression, and recent breakthroughs in the understanding of translation are driving the design of new therapeutic strategies for RP-mutant cancer [5]. Ideally, one would therapeutically target only the mutant ribosomes without affecting wild-type ribosomes. Many existing antibiotics bind and inhibit prokaryotic ribosomes. Recent structural analysis of prokaryotic and eukaryotic ribosomes demonstrate that selectivity of antibiotics binding is often provided by very subtle differences in ribosomal structure [12,88]. Moreover, resistance to antibiotics due to impaired binding to the ribosome often relies on changes as faint as a single chemical modification of the ribosome [12,89]. These observations suggest that it is feasible to develop small molecules that target RP-mutant cancer ribosomes, provided that these have structural features that distinguish them from ribosomes in healthy cells. In the absence of agents that directly target the mutant ribosomes, one can instead target the downstream consequences of the mutant ribosomes. In this regard, a proof-of-concept was recently presented in the context of the leukemia-associated *RPL10-R98S* (uL16-R98S) mutation. RPL10-R98S (uL16-R98S) ribosomes were shown to drive constitutive translation of the survival protein BCL-2. As a consequence, leukemias carrying the RPL10-R98S (uL16-R98S) defect are highly sensitive to the clinically used BCL-2 inhibitor Venetoclax [37]. Similarly, therapeutic approaches targeting cancer cells with other somatic RP deletions or mutations and the associated proteome changes may be identified. However, the congenital ribosomal defects as seen in ribosomopathies are much more difficult to address. Therapeutic compounds that could selectively target these cells are unfortunately not useful, as these patients carry the ribosome defect in each cell of their body. Such agents would thus target all cells, including those not giving rise to cancer and essential to sustain life. A more specific therapeutic approach should therefore be designed. For instance, the RP-defective cancer cell targeting compounds could be conjugated to antibodies against tumor-specific antigens. The success of such an approach however depends on availability of such antigens. Moreover, this strategy would not correct the RP defect in the non-cancerous cells in the body, and it may just be a matter of time before one of these remaining cells transitions from hypo- to hyperproliferation and gives rise to another cancer. 

Administration of L-leucine, a known activator of mRNA translation, has shown promising results for correcting the anemia and developmental defects in DBA and 5q- syndrome [90,91]. It is however unclear if L-leucine is also able to reduce the risk of cancer transition associated with these diseases. Although it dampens activation of Tp53 target genes in a DBA Rps19 (eS19) mouse model [91], L-leucine does not impair the ribosomal stress–induced Tp53 response in *Rps19* (eS19) and *Rps14* (uS11) morpholino zebrafish and human CD34+ cells [92]. An intact TP53 response upon administration of L-leucine may thus protect the cells from transformation. However, because L-leucine again does not correct the actual ribosomal defect in these diseases, it might still promote usage of defective ribosomes. It is currently unclear whether translational fidelity is perturbed in DBA and MDS. If this is the case, combining L-leucine with drugs that correct fidelity defects may be required. Such classes of drugs are an active field of research. For example, drug screens have identified compounds that decrease the fidelity of start codon initiation [93]. Moreover, ataluren (Translarna), a drug promoting premature stop-codon read-through [94], is in clinical trials for treatment of diseases caused by nonsense mutations, such as Duchenne muscular dystrophy and cystic fibrosis [95,96]. Because stop-codon read-through has recently emerged as a relevant anti-angiogenic mechanism [97], similar drugs may also find applications in cancer and ribosomopathy treatments in the future.

Finally, immunotherapy is one of the most exciting recent breakthroughs in cancer therapy, and might be a particularly promising treatment option for RP-mutant patients. Tumor mutation load was recently defined as a biomarker of response to checkpoint inhibitor immunotherapy, with higher mutational burdens corresponding to improved response rates in a variety of cancers [98,99,100]. If tumors in patients with congenital ribosome defects carry a higher mutational load, as recently observed in somatic RP-mutant cancer patients [69], they could benefit from such targeted therapy in the future. 

## Figures and Tables

**Figure 1 cells-08-00229-f001:**
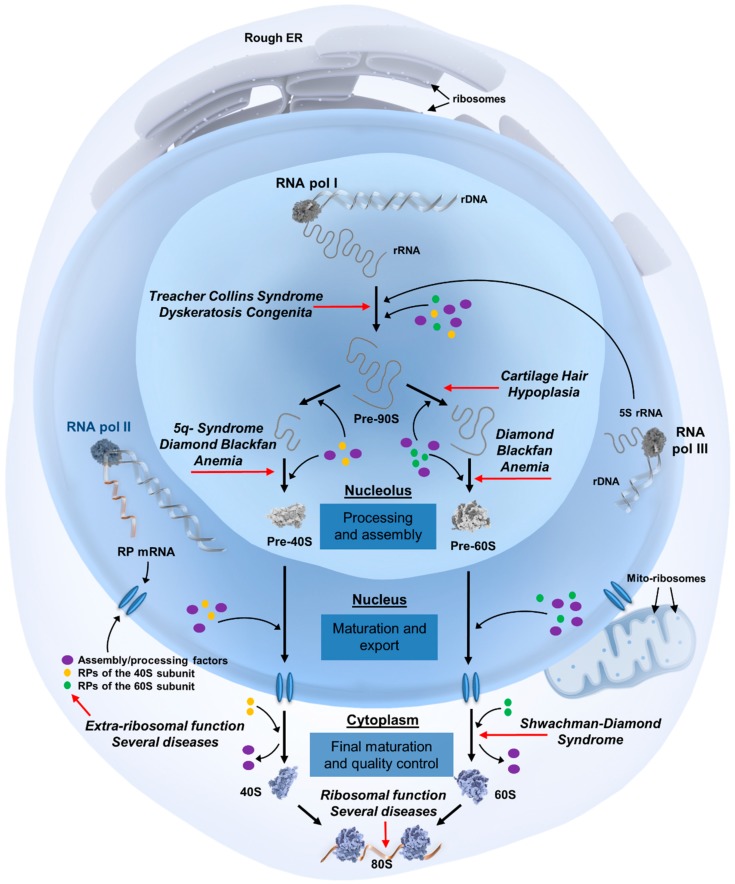
Overview of ribosome biogenesis and the steps affected in ribosomopathies. The eukaryotic ribosome assembly process begins in the nucleolus, a sub-compartment of the nucleus that is organized around ribosomal DNA (rDNA) transcription units. Three of the four main ribosomal RNA (rRNA) molecules are transcribed in the nucleolus as precursor rRNA by RNA polymerase I, with the last rRNA molecule being transcribed in the nucleoplasm by RNA polymerase III. These pre-rRNAs undergo extensive structural modifications before becoming the final 18S, 25S, 5.8S, and 5S rRNAs. Ribosomal protein mRNAs are transcribed by RNA polymerase II in the nucleoplasm, translated into proteins in the cytoplasm, after which they are transported back to the nucleolus to participate in additional processing steps. These steps comprise the formation of numerous assembly intermediates, including a 90S pre-ribosome that begins to fold and further associate with ribosomal proteins, cleavage steps resulting in 43S and 66S pre-ribosomal particles, and their transport from the nucleolus into the nucleus and subsequently the cytoplasm for final rRNA processing and protein assembly events. Both subunits are exported into the cytoplasm with a complement of non-ribosomal factors, some of which facilitate export while others prevent the premature interaction of the ribosomal subunits. These factors must be released in the cytoplasm and shuttled back to the nucleus for subsequent rounds of maturation and export. After this recycling step, which is frequently coupled to final structural and functional quality controls of the assembled subunits, mature large 60S and small 40S subunits can interact to form translationally active 80S ribosomes. For simplicity, trans-acting biogenesis factors are referred to as assembly/processing factors, and can include ATP/GTPases, helicases, nucleases, kinases, small nucleolar RNAs (snoRNAs), chaperones, and protein co-factors. The biogenesis steps affected in ribosomopathies are marked with red arrows. Abbreviations: ER: endoplasmic reticulum; pol: polymerase; RP: ribosomal protein; mito-ribosomes: mitochondrial ribosomes.

**Figure 2 cells-08-00229-f002:**
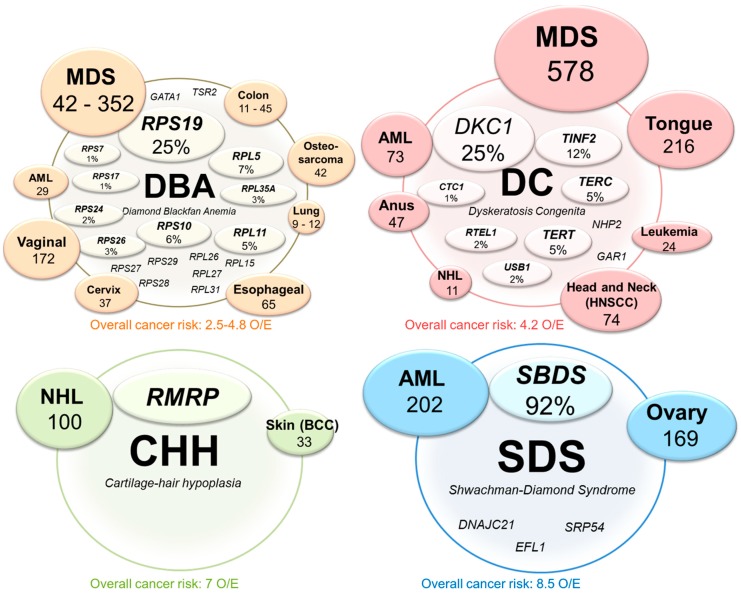
Overview of the most common mutations and associated cancer risks in congenital ribosomopathies. The genes that are mutated or deleted in Diamond Blackfan anemia (DBA), dyskeratosis congenita (DC), cartilage-hair hypoplasia (CHH) and Shwachman–Diamond Syndrome (SDS) are indicated in the lightly shaded inner circles. The reported percentages indicate the fraction of patients with that ribosomopathy that presents a genetic defect in the corresponding gene. When the exact incidence of the defects are unknown, only the gene name is indicated. The outer, darker circles report the cancer types that occur at a significantly higher incidence in the shown ribosomopathy as compared to the healthy population. The size of the circles and the number reported below each cancer type represents the observed over expected (O/E) ratio and quantifies the risk increase for that particular cancer type in each ribosomopathy. Below the circle of each ribosomopathy, the overall cancer risk (O/E ratio) is reported.

**Figure 3 cells-08-00229-f003:**
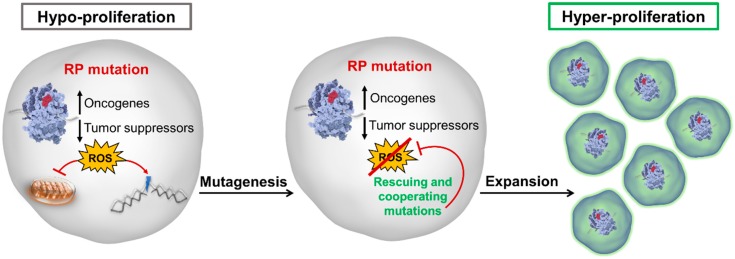
A model of the cellular transition from hypo- to hyperproliferation in ribosome-mutant diseases. Ribosomal defects alter the cellular translational landscape, promoting a protein expression profile that favors growth-promoting and disfavors tumor-suppressing factors. Defects in the assembly and/or function of ribosomes also leads to increased cellular stress and production of reactive oxygen species (ROS), which inhibits cellular proliferation and promotes DNA damage and higher mutagenesis. Over time, rescuing mutations arise that inhibit ROS production, thereby removing the block on cellular proliferation. This unleashes the cellular oncogenic potential, marking the beginning of the affected cell’s progression to transformation. RP: ribosomal protein.

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
