# Peer review of "Cancer Biogenesis in Ribosomopathies"

_cells, 2019, doi:10.3390/cells8030229_

Round 1

Reviewer 1 Report

In this review manuscript Sulima and colleagues discuss specifically cancer susceptibility in ribosomopathies. Despite a number of reviews on ribosomopaties are available this particular aspect deserves the attention of readers in a dedicated review article. The authors deal with this topic with a very broad and balanced approach.

The manuscript is well organised and requires in my opinion minor corrections/specifications:

The authors use the classical nomenclature for ribosomal proteins and this is understandable considering the manuscript is for non specialised audince. 

Sometimes they however also indicate the new nomenclature (e.g. line 216, line 227), this should be made homogeneous through the text.

Figure 1 legend should include a list of abbreviations to be considered as a stand-alone element of the manuscript (e.g. ER, RP, etc.)

Also, while Treacher Collins syndrome has consequences mainly at the level of rRNA transcription, rRNA pseudouridylation that's found altered in dyskeratosis occurs both co-transcriptionally and post-transcriptionally, this is not evident from the image.

Additionally, I find confounding showing that PolII transcribes rRNA, please add 5S rRNA or 5S RNA genes.

Figure 2

Overall cancer incidence (or relative risk) should be indicated for the 4 diseases in the picture.

Autosomal recessive DC is caused also by mutations in genes encoding for NHP2 & GAR1, which participate in the pseudouridylation complex, please indicate it

Author Response

In this review manuscript Sulima and colleagues discuss specifically cancer susceptibility in ribosomopathies. Despite a number of reviews on ribosomopathies are available this particular aspect deserves the attention of readers in a dedicated review article. The authors deal with this topic with a very broad and balanced approach. The manuscript is well organized and requires in my opinion minor corrections/specifications:

·         The authors use the classical nomenclature for ribosomal proteins and this is understandable considering the manuscript is for non-specialized audience. 

                Sometimes they however also indicate the new nomenclature (e.g. line 216, line 227), this should be                made homogeneous through the text.

                Answer: This has now been made homogeneous throughout the text.

·         Figure 1 legend should include a list of abbreviations to be considered as a stand-alone element of the manuscript (e.g. ER, RP, etc.)

Answer: A list of abbreviations has been added to Figure legend 1.

·         Also, while Treacher Collins syndrome has consequences mainly at the level of rRNA transcription, rRNA pseudouridylation that's found altered in dyskeratosis occurs both co-transcriptionally and post-transcriptionally, this is not evident from the image.

Answer: We generated Figure 1 to provide a general, broad picture. It is certainly a simplification, and it would be impossible to show all the details properly in one such figure. Pseudouridylation is not singled out in this figure, but for simplicity falls under the category of “assembly and processing factors”, which includes many other proteins and snoRNAs (this is also commented on in the figure legend). Therefore, it is not possible to address this concern. However, pseudouridylation is discussed in some detail in the text.

·         Additionally, I find confounding showing that PolII transcribes rRNA, please add 5S rRNA or 5S RNA genes.

Answer: This has been fixed.

·         Figure 2: Overall cancer incidence (or relative risk) should be indicated for the 4 diseases in the picture.

Answer: Overall cancer risks (observed/expected ratios) have been added in Figure 2.

·         Figure 2: Autosomal recessive DC is caused also by mutations in genes encoding for NHP2 & GAR1, which participate in the pseudouridylation complex, please indicate it

Answer: NHP2 and GAR1 have been added to the Figure.

Reviewer 2 Report

The review submitted by Sulima and colleagues titled "Cancer Biogenesis in Ribosomopathies" offers a perspective on the effects of altered ribosome function due to mutated ribosomal proteins on the growth and proliferation of cells.  It provides a solid foundation of basic ribosome biogenesis and the role of the ribosome in cellular proliferation.  Focusing on mutations found in a group of diseases collectively known as ribosomopathies, the authors provide a link between these diseases and later cancer development.   Data pointing to potential mechanisms of the switch from the ribosomopathic diseases characterized by defects in cellular growth and proliferation to tumorigenesis is discussed.  Finally, potential therapeutic targets for intervention in the cancers associated with ribosomal defects are summarized.  Overall the review is logically and clearly presented.

Main concerns

It is not clear from the review in the section Category III: Cellular stress and metabolic states, what the mechanism of mutant/misassembled ribosome-driven ROS production is.  A short discussion of what is known about how ROS is produced as a result of defective ribosome production at the beginning of this section would help the reader better understand the subsequent results of ROS on cellular growth and proliferation.

One additional mechanism of ribosomopathy driven hyperproliferation which should be discussed is the direct effect of ROS production on cellular proliferation.  There are several studies demonstrating that ROS production  itself can directly increase cellular proliferation through activation of progrowth pathways such as the MAPK pathway.

Minor points

1 - Reference # 74 is incomplete requiring the journal name HemaSphere to be added.

2 - The color of the rRNA depicted in the Nucleolus in Figure 1 is difficult to see.  It blends into the background.

3 - Reference 78 needs to be removed from the manuscript unless it is now "in press."  Manuscripts under review are not valid references.

Author Response

The review submitted by Sulima and colleagues titled "Cancer Biogenesis in Ribosomopathies" offers a perspective on the effects of altered ribosome function due to mutated ribosomal proteins on the growth and proliferation of cells.  It provides a solid foundation of basic ribosome biogenesis and the role of the ribosome in cellular proliferation.  Focusing on mutations found in a group of diseases collectively known as ribosomopathies, the authors provide a link between these diseases and later cancer development.  Data pointing to potential mechanisms of the switch from the ribosomopathic diseases characterized by defects in cellular growth and proliferation to tumorigenesis is discussed.  Finally, potential therapeutic targets for intervention in the cancers associated with ribosomal defects are summarized.  Overall the review is logically and clearly presented.

We thank the reviewer for the nice comments on our review article.

Main concerns

•             It is not clear from the review in the section Category III: Cellular stress and metabolic states, what the mechanism of mutant/misassembled ribosome-driven ROS production is.  A short discussion of what is known about how ROS is produced as a result of defective ribosome production at the beginning of this section would help the reader better understand the subsequent results of ROS on cellular growth and proliferation.

 Answer: We had briefly touched on this point at the end of the first paragraph of this section and have now added additional information by adding the following text and the associated refs on lines 323-335: ‘ The mechanisms by which defective ribosomes lead to elevated ROS are poorly understood. In the case of the leukemia associated RPL10-R98S (uL16-R98S) mutation, enhanced levels of ROS may arise from increased peroxisome activity. Peroxisomes are cellular organelles in which oxidation of long-chain fatty acids occurs, resulting in the production of high levels of hydrogen peroxide (H2O2). Several peroxisomal enzymes, such as PAOX, are transcriptionally upregulated in RPL10-R98S (uL16-R98S) cells. By what mechanisms mutant ribosomes drive peroxisomal oxidation is unclear. Wild type RPL10 (uL16) has also been described to regulate the expression of proteins related to ROS production and to control mitochondrial ROS production in pancreatic cancer. It is therefore possible that RPL10-R98S (uL16-R98S) is no longer able to properly perform these ROS regulatory functions, but this requires further research. Other ribosomal proteins have also been involved in oxidative stress responses. For example, ROS-inducing agents can cause RPS3 (uS3) to translocate to the mitochondria, where it can protect the cells from ROS-induced mitochondrial DNA damage.’

•             One additional mechanism of ribosomopathy driven hyperproliferation which should be discussed is the direct effect of ROS production on cellular proliferation.  There are several studies demonstrating that ROS production itself can directly increase cellular proliferation through activation of progrowth pathways such as the MAPK pathway.

Answer: Elevation of ROS levels can indeed initiate different cellular outcomes, depending on the levels that are achieved. At low levels, cellular ROS levels can stimulate proliferation by activating the PI3K and MAPK signaling pathways. At higher levels, ROS associated oxidative stress becomes toxic and rather inhibits proliferation. In the context of ribosomal protein mutations, the latter seems to be the case: reduction of cellular ROS levels by means of an anti-oxidant can rescue the proliferation defects in RPL10-R98S cells, supporting that RPL10-R98S associated ROS production impairs cell proliferation. This text and the accompanying references have been added on lines 336-342.

Minor points

•             Reference # 74 is incomplete requiring the journal name HemaSphere to be added.

Answer: The journal name has been added.

•             The color of the rRNA depicted in the Nucleolus in Figure 1 is difficult to see.  It blends into the background.

Answer: This has been addressed by changing the color of the nucleus to a lighter shade.

•             Reference 78 needs to be removed from the manuscript unless it is now "in press."  Manuscripts under review are not valid references.

Answer: We had anticipated that reviewing and revision of this Cells review would take much longer and that our manuscript described in ref 78 (now ref 85 after revision) would have been accepted by the time we would resubmit the revision of this review article. Everything however went extremely fast for this review and we only have 5 days to perform the revision. We requested an extension to the Cells editor to allow the final decision on reference 78 to be done but the editor preferred not to delay resubmission of our revised Cells manuscript and preferred that we keep reference 78 as it is. The Cells journal policy is such that one can refer to unpublished material in the text as we did.

Reviewer 3 Report

In this review, Sulima et al. provide an updated vision about the current evidence and mechanistic models underlying the link between ribosome biogenesis defects and cancer.

It is an excellent review that extends and complements other ones on the same topic. One marked aspect is the comprehensive integration of data on cancer risk in ribosomopathies in quantitative terms. Another informative section is the one on oncogenic mechanisms. In this part, the authors not only update and extend the description of previously-proposed models, such as the alterations in ribosome translation, but pose new ideas on the possible implications of cellular stress and metabolic states in the development of cancer. 

The text is very well-written and nicely structured. 

Minor points:

1.     Page 2, line 55. Include, as and additional reference, the recent review by Aspesi and Ellis (Nat. Rev. Cell Biol, 2019)

2.     Page 10, line 403. Update reference 78.

Author Response

In this review, Sulima et al. provide an updated vision about the current evidence and mechanistic models underlying the link between ribosome biogenesis defects and cancer. It is an excellent review that extends and complements other ones on the same topic. One marked aspect is the comprehensive integration of data on cancer risk in ribosomopathies in quantitative terms. Another informative section is the one on oncogenic mechanisms. In this part, the authors not only update and extend the description of previously-proposed models, such as the alterations in ribosome translation, but pose new ideas on the possible implications of cellular stress and metabolic states in the development of cancer. The text is very well-written and nicely structured. 

We thank the reviewer for the nice comments on our review article.

Minor points:

·         Page 2, line 55. Include, as and additional reference, the recent review by Aspesi and Ellis (Nat. Rev. Cell Biol, 2019)

Answer: We have added the requested reference.

·         Page 10, line 403. Update reference 78.

                Answer: We have updated this ref (now ref 85 after revision) with its current status. We                     were in contact with the Cells editor about this and the journal policy is that one can refer                 to unpublished material in the text as we did.